# Aerobic Degradation Characteristics and Mechanism of Decabromodiphenyl Ether (BDE-209) Using Complex Bacteria Communities

**DOI:** 10.3390/ijerph192417012

**Published:** 2022-12-18

**Authors:** Dingfan Hu, Juan Wu, Luosheng Fan, Shunyao Li, Rong Jia

**Affiliations:** 1School of Resources and Environmental Engineering, Anhui University, Hefei 230601, China; 2School of Life Sciences, Anhui University, Hefei 230601, China

**Keywords:** decabromodiphenyl ether, complex bacteria community, degradation efficiency, pathway, crude enzyme, cytochrome P450

## Abstract

Complex bacteria communities that comprised *Brevibacillus* sp. (M1) and *Achromobacter* sp. (M2) with effective abilities of degrading decabromodiphenyl ether (BDE-209) were investigated for their degradation characteristics and mechanisms under aerobic conditions. The experimental results indicated that 88.4% of 10 mg L^−1^ BDE-209 could be degraded after incubation for 120 h under the optimum conditions of pH 7.0, 30 °C and 15% of the inoculation volume, and the addition ratio of two bacterial suspensions was 1:1. Based on the identification of BDE-209 degradation products via liquid chromatography–mass spectrometry (LC–MS) analysis, the biodegradation pathway of BDE-209 was proposed. The debromination, hydroxylation, deprotonation, breakage of ether bonds and ring-opening processes were included in the degradation process. Furthermore, intracellular enzymes had the greatest contribution to BDE-209 biodegradation, and the inhibition of piperyl butoxide (PB) for BDE-209 degradation revealed that the cytochrome P450 (CYP) enzyme was likely the key enzyme during BDE-209 degradation by bacteria M (1+2). Our study provided alternative ideas for the microbial degradation of BDE-209 by aerobic complex bacteria communities in a water system.

## 1. Introduction

As a brominated flame retardant with excellent performance, decabromodiphenyl ether (BDE-209) is extensively applied to electronics, electrical appliances, plastics, furniture and textiles [1,2,3] owing to its high flame retardant properties, good stability to heat and low cost. Due to its long half-life and strong lipophilicity, BDE-209 has been widely detected in water, air, sediment and soil [4,5,6]. BDE-209 has the tendency to be biomagnified along the food chain [7,8,9]. Toxicological studies have indicated that BDE-209 has liver toxicity [10,11], reproductive toxicity [12,13], neurotoxicity [14,15] and immunotoxicity [16] and can cause mitochondrial dysfunction [17]. On account of its lipophilicity and toxicity, BDE-209 has exhibited serious impacts on human health and ecological safety. Therefore, BDE-209 contamination has attracted widespread attention, and it is of great importance to enhance BDE-209 degradation technology.

At present, some technologies have been developed to remove BDE-209 from the environment, including chemical, physical and biological methods [18,19,20]. However, these physical or chemical methods still have some problems that have not been well solved. For example, photocatalytic reduction, nano-zero valent iron and iron-based bimetallic materials have been used for the elimination of BDE-209 [21,22,23]; however, most of the reduction processes were incomplete and could produce low-brominated poly brominated diphenyl ethers (PBDEs) congeners, which were even more toxic and difficult to be degraded. As an effective and environmentally friendly approach, biodegradation has been considered a promising technology and one of the most ideal methods for BDE-209 in situ remediation. According to some research, BDE-209 can be degraded by anaerobic or aerobic microorganisms. Shi et al. performed anaerobic digestion of BDE-209 for 210 d in a thermophilic continuously stirred anaerobic bioreactor and found that the maximum removal rate of BDE-209 was 1.1 μg·day^−1^ [24]. However, compared with anaerobic degradation [25,26,27], the aerobic degradation process typically took a shorter period, and probably resulted in complete mineralization [28,29,30]. Paliya et al. isolated a novel bacterium, *Bacillus tequilensis*, from a municipal landfill, which could degrade 65% of 50 mg L^−1^ BDE-209 within 8 d aerobically [31]. Moreover, it has been reported that degrading enzymes including various oxygenases secreted by aerobic bacteria were responsible for benzene ring lysis and the complete mineralization of BDE-209 [32,33,34].

Although a lot of studies on BDE-209 biodegradation have been carried out, bio-enhanced repair using a single degrading strain in the actual environment did not result in ideal effects. The main problem was that a single degrading strain could not cope well with environmental disturbances, resulting in a low survival rate, leading to the ideal state on which bioremediation relied [35]. There have been some reports on the efficient degradation of residual pesticides based on microbial consortiums [36,37,38,39]. Zhang et al. used *Diaphorobacter* sp. LR2014-1 and *Achromobacter* sp. ANB-1 strains isolated from a linuron-mineralizing consortium to study the degradation of linuron. It was found that the co-catabolism of the two strains for linuron resulted in higher catabolism efficiency and better growth of both strains [40]. Yu et al. isolated a novel microbial consortium, GY1, from e-waste-polluted farmland and found that it could effectively promote BDE-209 degradation in actual water–sediment systems [41]. Although complex bacteria communities have been considered a promising option in bioremediation, studies on BDE-209 biodegradation by aerobic bacteria consortiums are still limited due to the lack of efficient bacteria and the complexity of the species [42,43]. On account of the urgent desire for bioremediation, it has practical significance to explore the application of complex bacteria communities in BDE-209 biodegradation.

Up until now, there have been few studies on the influences of BDE-209 on complex bacteria communities, including the bioremediation abilities as well as the biodegradation mechanisms of complex bacteria communities. In addition, the role of oxygenase in the BDE-209 biodegradation process using complex bacteria communities has seldom been studied. In our previous research, some bacteria that could use BDE-209 as the sole carbon source were isolated from activated sludge [44]. In order to reveal the degradation characteristics of these strains for BDE-209 thoroughly, four of these strains were selected to carry out further studies in this research. The biodegradation capabilities of different complex bacteria communities for BDE-209 were investigated first, and the factors affecting BDE-209 biodegradation by multiple microorganisms were examined. In order to analyze the biodegradation mechanisms of BDE-209 by complex bacteria communities, the metabolites were identified via liquid chromatography–mass spectrometry (LC–MS), and the degradation pathway of BDE-209 was revealed. Furthermore, the degradation of BDE-209 by crude enzymes and the influence of enzyme inhibitors on BDE-209 biodegradation were also explored. This research is of great importance in improving the biodegradability of microorganisms for BDE-209 and developing complex bacterial community remediation technology in the environment contaminated by brominated flame retardants (BFRs).

## 2. Materials and Methods

### 2.1. Chemicals and Reagents

BDE-209 (98% purity) used in this work was acquired from Aladdin industrial companies. High-performance liquid chromatography (HPLC)-grade Dimethyl sulfoxide, dichloromethane, n-hexane, tetrahydrofuran acetonitrile and piperyl butoxide (PB, 95% purity) were obtained from Macklin industrial companies. Other analytical grade reagents were provided by Sinopharm Chemical Reagent Company (Shanghai, China).

### 2.2. Microorganisms and Culture Medium

The four strains used in the experiments were screened and purified from activated sludge from the Hefei Wang Xiaoying sewage treatment plant in China in our previous research [44].

Luria–Bertani (LB) medium was applied to strain culture, which consisted of (g L^−1^): pepton 10.0, yeast leaching powder 5.0 and NaCl 5.0, and pH was adjusted to 7.2. Mineral salt medium (MSM) was used as the degradation medium, and contained (g L^−1^): KH_2_PO_4_ 2.65, NaHPO_4_·12H_2_O 4.26, NaCl 5 and NH_4_Cl 5, and 1 mL of trace elements solution, with pH adjusted to 7.0. The trace elements solution was prepared using (g L^−1^): FeSO_4_·7H_2_O 1.0, ZnCl_2_ 1.0, MgSO_4_·7H_2_O 1.0, CuSO_4_·5H_2_O 1.0, MnSO_4_·H_2_O 1.0, CaCl_2_ 1.0 and CoCl_2_·6H_2_O 0.2.

### 2.3. Preparation of Bacterial Suspension

The single strain was inoculated in 50 mL LB medium and incubated at 30 °C, 150 rpm for 24 h. Whereafter, the cells were gathered via centrifugation at 4 °C, 8000× *g* for 4 min, and then rinsed three times using MSM. Finally, the harvested cells were resuspended in MSM to prepare bacterial suspension of single strain, and the cell density was about OD_600_ 1.0. The bacterial suspension of complex bacteria communities used in degradation experiments was prepared by mixing different bacterial suspensions from single strain in equal proportions. The inoculation amount was 15% (*V/V*) in all degradation experiments.

### 2.4. BDE-209 Biodegradation Experiments

BDE-209 was dissolved in dimethyl sulfoxide to give a 1000 mg L^−1^ BDE-209 solution. Next, 200 μL of above BDE-209 solution was added to sterilized MSM so that the initial concentration of BDE-209 in degradation system was 10 mg L^−1^. Then 3 mL bacterial suspension was inoculated into the medium, making the total volume of degradation system 20 mL. BDE-209 was biodegraded using single strain and microbial consortiums of two, three and four strains. The degradation experiments were carried out at 30 °C, shaken on a rotary shaker at 150 rpm for 120 h in darkness. Control experiments were performed in MSM containing 10 mg L^−1^ BDE-209 without inoculating bacteria. All experiments were conducted in triplicate. The residual concentration of BDE-209 in MSM was determined to calculate the biodegradation efficiency of BDE-209. The degradation efficiency of BDE-209 was calculated according to the following formula:Degradation efficiency (%) = (C_0_ − C_t_)/C_0_

C_0_ and C_t_ represent the initial and final concentrations of BDE-209, respectively.

### 2.5. Optimization of Degradation Conditions

The optimization experiments were carried out using the one-factor variable method, and the selected three conditions were temperature, pH value and inoculation volume. The biodegradation experiments were carried out as described before with BDE-209 initial concentration of 10 mg L^−1^ for 120 h at 150 rpm in the dark. The optimization experiments were designed as shown in Table 1. When temperature was changed, pH and inoculation volume remained at 7.0 and 15%, respectively. When pH was changed, temperature and inoculation volume remained at 30 °C and 15%, respectively. When inoculation volume was changed, pH and temperature remained at 7.0 and 30 °C, respectively. The biodegradation process of BDE-209 with time was also investigated under the optimal degradation conditions, and samples were taken at 6, 12, 24, 72 and 120 h, respectively.

### 2.6. Biodegradation of BDE-209 by Crude Enzymes

#### 2.6.1. Preparation of Extracellular and Intracellular Crude Enzymes

The strain used in the experiments was cultured in LB medium for 24 h. Then the culture medium was centrifuged at 4 °C, 8000× *g* for 4 min, and the supernatant and cells were collected separately. The supernatant was filtered through 0.22 μm filter membrane, and the filtrate was used as extracellular crude enzyme solution. After the cells were rinsed three times with phosphate-buffered solution (PBS), they were suspended in PBS again and placed in 2 mL centrifuge tubes with grinding beads. Then, the above solution was placed in a bead mill to be broken at a frequency of 50 Hz for 180 s. Next, the mixture was centrifuged at 8000× *g*, 4 °C for 5 min to remove cell debris, and then the resultant supernatant was filtered with 0.22 μm filter membrane. Finally, the produced filtrate was used as intracellular crude enzyme solution.

#### 2.6.2. Degradation Experiments of BDE-209 by Crude Enzymes

In order to obtain a degradation system with BDE-209 initial concentration of 10 mg L^−1^, 20 μL of 1000 mg L^−1^ BDE-209 stock solution was transferred into the intracellular and extracellular crude enzyme solutions prepared as described above, and the total volume of both reaction systems was 2 mL. The biodegradation experiments of BDE-209 by extracellular and intracellular crude enzymes were carried out in four different ways: using extracellular crude enzyme liquid of single bacterium, using extracellular crude enzyme liquid of complex bacteria community, using intracellular crude enzyme liquid of single bacterium, and using intracellular crude enzyme liquid of complex bacteria community. The degradation experiments were conducted at 150 rpm, 30 °C for 24 h in darkness. Control experiments were performed in PBS containing 10 mg L^−1^ BDE-209 without crude enzyme solution.

### 2.7. Cytochrome P450 Enzyme Inhibitor Experiments

To verify the function of cytochrome P450 (CYP450) monooxygenases in BDE-209 biodegradation, piperyl butoxide (PB), the universal CYP450 inhibitor, was added to MSM and pre-incubated with complex bacteria community for 2 h, with initial PB levels of 1, 2, 5, and 20 mmol L^−1^. Then BDE-209 was added to the above system (10 mg L^−1^) to be degraded for another 120 h. The degradation system without PB was used as control.

### 2.8. Extraction of BDE-209 after Biodegradation

During degradation experiments, BDE-209 might be adsorbed on the surface of microorganisms due to its strong hydrophobicity. To avoid the loss of BDE-209, the entirety of the bottle medium (20 mL) was extracted at desired time. A total of 20 mL of culture medium was extracted with equal volumes of dichloromethane and n-hexane (1:1, *V/V*) in 125 mL separating funnel, and the organic phase was collected after dehydration with anhydrous sodium sulfate. Then, the organic phase was concentrated to 1 mL through rotary evaporation at 120 rpm and 40 °C, and the residue produced that was stuck to the inner wall of bottle was washed with tetrahydrofuran and fixed to 10 mL. Next, 1 mL of above volume solution was taken and filtered through an organic membrane of 0.22 μm and stored in brown injection vial.

The residual concentration of BDE-209 was analyzed via HPLC (Waters Alliance E2695), and the intermediates during BDE-209 degradation were monitored and identified via LC (Thermo SCIENTIFIC HPLC, Waltham, MA, USA)–MS (LTQ ORBITRAP XL). The detailed experimental methods are presented in Appendix A.

The same extraction procedure with BDE-209 was used to analyze the degradation products of BDE-209 through LC–MS in this paper.

### 2.9. Quality Assurance and Quality Control (QA/QC)

BDE-209 was analyzed via HPLC, the method detection limit was 0.0167 mg/L, and the limit of quantitation was 0.05 mg/L. The recoveries of the analyte were in the range of 88.0% to 113.4% (mean value 99.9%). The relative standard deviation (RSD) of the analyte recoveries ranged from 0.7% to 2.9%.

### 2.10. Statistical Analysis

The experiments in this paper were performed in triplicate, and all of the data appearing in the text are mean values of three independent replicate treatments. OriginPro 2021 (9.8.0.200) was used to conduct statistical analysis and data processing, and the phylogenetic trees were built with MEGA X (10.1.7). IBM SPSS Statistics 26.0 was used to analyze statistical differences among different treatment groups. A probability of *p* < 0.05 was considered statistically significant.

## 3. Results

### 3.1. Identification of Aerobic Degrading Bacteria for BDE-209

The identification of the four strains used in this paper was completed by Bioengineering Co., Ltd. (Shanghai, China). The phylogenetic trees were constructed using a 16S Rrna gene sequence (Figure 1). By comparing with data available in the Gen-bank database, the results showed that strain M1 shared 100.00% similarity with *Brevibacillus* sp. N3, strain M2 shared 99.65% similarity with *Achromobacter xylosoxidans* BN2910, strain M3 shared 99.79% similarity with *Achromobacter xylosoxidans* CD-253, and strain M4 shared 99.93% similarity with *Pseudomonas denitrificans* ATCC 13867. Thus, strains M1, M2, M3 and M4 were preliminarily identified as *Brevibacillus* sp., *Achromobacter xylosoxidans*, *Achromobacter xylosoxidans* and *Pseudomonas denitrificans*, respectively.

### 3.2. BDE-209 Biodegradation by Complex Bacteria Communities

Although the selected four strains could utilize BDE-209 as the only carbon source, an obvious difference in degradation ability could be observed. As seen in Figure 2, strain M4 shows the best degradation effect for BDE-209 among the four single strains. Compared with the results of BDE-209 degradation by a single bacterium, the degradation effect of complex bacteria communities for BDE-209 was higher except for M (1+4) and M (2+3). Among the combinations of two strains, the highest degradation efficiency of 81.3% was obtained by the combination of strains M1 and strain M2; however, the combination of strain M2 and strain M3 showed the worst degradation effect for BDE-209. The fact that there were synergistic, additive or antagonistic effects between different bacteria could be used to explain the above result. It was also found that the combination of three strains and four strains generally showed great BDE-209 degradation ability, and the minimum degradation efficiency could reach 62.4% higher than that of a single strain. In the current study, BDE-209 could be degraded more effectively by some complex bacteria communities than by a single bacterium. The complex bacteria community M (1+2), which had the highest degradation efficiency, was selected as the strains for further experimentation.

### 3.3. Factors Affecting BDE-209 Biodegradation by Complex Bacteria Community M (1+2)

The temperature, pH values of the degradation system and inoculation volume were important factors affecting the biodegradation of BDE-209 because the synthesis of degradation enzymes and the growth of microorganisms were significantly influenced by these factors. Therefore, optimization experiments were conducted to improve the biodegradability of BDE-209 by the complex bacteria community M (1+2). It can be seen from Figure 3a that there was no obvious change in degradation efficiency when the temperature was in the range of 10~25 °C. However, when the temperature reached 30 °C, the degradation efficiency of BDE-209 was 88.4%, which was higher than that at other temperatures. The degradation efficiency of BDE-209 declined to 14.0% at the higher temperature of 40 °C. The results shown in Figure 3b reveal that the degradation efficiency of BDE-209 varied from 69.3–85.6% for 10 mg L^−1^ BDE-209. The degradation efficiency was almost the same between pH 6.0 and 8.0, and more than 80% of BDE-209 was degraded. However, the degradation efficiency decreased by 14.5% at pH 5.0 and by 11% at pH 9.0 compared with that of pH 7.0.

The experimental results of BDE-209 degradation under different inoculation volumes are shown in Figure 3c. The degradation efficiency of BDE-209 was obviously improved and reached more than 80% when the inoculation volume was higher than 1%.

In addition to the above three conditions, the addition ratio of bacterial suspension of M1 and M2 was also investigated under the conditions of pH 7.0, 30 °C and 15% of inoculation volume. As seen in Figure 3d, the addition ratio of bacterial suspension of M1 and M2 had little effect on the degradation efficiency of BDE-209, and the degradation efficiency of BDE-209 varied from 83.7–88.4%, which showed effective biodegradation of BDE-209 within the test range of the ratio of bacterial suspension of M1 and M2. The biodegradation rate of BDE-209 by complex bacteria community M (1+2) was fast, as shown in Figure 3e. The degradation efficiency reached 64.1% at 6 h, and after that increased gradually with time to 87.8% at 120 h.

### 3.4. Analysis of Metabolites and Pathway for BDE-209 Biodegradation

To clarify the biodegradation mechanism of BDE-209 by complex bacteria community M (1+2), the metabolites of BDE-209 during degradation were detected and identified through LC–MS analysis. The samples were taken at 24 h, 72 h and 120 h during the experiments. In the process of degradation of BDE-209, a total of 11 metabolites were detected. The mass spectra and chemical structures of identified intermediates are presented in Appendix A and Table 2, respectively. The degradation products were qualitatively detected by comparing their mass spectra to those in previous literature studies.

Based on the identified products and related literature, the possible biodegradation path for BDE-209 by complex bacteria community M (1+2) was proposed (Figure 4). During the aerobic degradation process, BDE-209 first underwent continuous debromination to form product (a), and then two possible biodegradation pathways occurred. In one pathway, product (a) was converted to product (b) by a hydroxylation reaction and was further decomposed into products (c) and (d) via diphenyl ether bond breakage. Subsequently, the resultant product (e) was attributed to the ring opening. In another degradation pathway, the hydroxylation reaction of product (a) resulted in the occurrence of product (f), which was then converted to product (g) or (h) through a deprotonation reaction. After that, product (g) was further metabolized to products (c) and (j), while product (h) was further transformed into products (c) and (i) by the breakage of the diphenyl ether bond and the ring opening. During the above process, product (c) could be first oxidized to product (d), and then product (d) could undergo ring-opening via ortho-cleavage. Finally, all the ring-opening products including products (e), (i), (j) and (k) might be further mineralized to CO_2_ and H_2_O by the tricarboxylic acid (TCA) cycle.

### 3.5. BDE-209 Biodegradation by Crude Enzymes

The results of BDE-209 degradation by extracellular and intracellular crude enzyme solutions are shown in Table 3, and the volume ratio of strain M1 and strain M2 crude enzyme solutions was 1:1. Not only for the single bacterium M1 but also for the complex bacteria community M (1+2), the extracellular crude enzymes had poor degradation effects for BDE-209. By contrast, intracellular crude enzymes were found to be more effective at degrading BDE-209 than extracellular enzymes, except for strain M2. At the same time, it was found that the degradation efficiency of complex bacteria community M (1+2) for BDE-209 was higher than that of any single strain using both extracellular crude enzymes and intracellular crude enzymes.

### 3.6. Effect of Cytochrome P450 Enzyme Inhibitor on BDE-209 Biodegradation

To examine whether the CYP450 monooxygenases participated in BDE-209 biodegradation by the complex bacteria community M (1+2), different concentrations of PB, a common inhibitor of CYP450, were added to the BDE-209 degradation system. As seen in Figure 5, the degradation efficiency of BDE-209 decreased significantly with the increasing concentration of the inhibitor PB. In the presence of 1, 2, 5 and 20 mmol L^−1^ PB, 50.7%, 39.0%, 30.7% and 31.3% of BDE-209 were degraded after 120 h of incubation, respectively. Compared with the control, the degradation efficiency decreased by 33.5% when the concentration of the inhibitor PB was 5 mmol L^−1^. The experimental results made it clear that the presence of PB decreased the degradation efficiency of BDE-209 by the complex bacteria community M (1+2) and that CYP might be responsible for BDE-209 biodegradation.

## 4. Discussion

At present, much research on BDE-209 biodegradation has been performed with anaerobic bacteria in soil or sediment. Nevertheless, the anaerobic degradation process usually takes a longer time, and the degradation efficiency is lower than the aerobic degradation process [24,26]. In our study, it was found that four aerobic bacteria could degrade BDE-209 to some extent in an aqueous solution. However, the unsatisfactory degradation effect of a single bacterium limited the application of microorganisms in bioremediation. Therefore, the degradation of BDE-209 by multiple microorganisms was explored in this study.

It was shown that BDE-209 aerobic degradation using complex bacteria communities is feasible. The combination of strain M1 and strain M2 showed higher degradation efficiency for BDE-209 than other complex bacteria communities, which suggested that there was a strong synergistic effect between *Brevibacillus* sp. (M1) and *Achromobacter* sp. (M2 and M3). Similarly, the synergistic effect between *Achromobacter* sp. (M2 and M3) and *Pseudomonas* sp. (M4) was also found in the experimental results. On the contrary, there were certain antagonistic effects between *Brevibacillus* sp. (M1) and *Pseudomonas* sp. (M4) and between the two *Achromobacter* sp. (M2 and M3). This was similar to the study reported by Liu et al. on the degradation of atrazine by bacterial communities [45]. Liu et al. found a significantly negative correlation between *Azospirillum* and *Halomonas* during the degradation of atrazine. Furthermore, the degradation efficiency of atrazine by the combination of *Arthrobacter* and *Methylophilus* was obviously higher than that of *Arthrobacter* only after inoculation for 48 h, and the ratio of relative abundance of the two strains had no influence on the degradation efficiency of atrazine [45]. Consequently, it is important to choose a proper combination of different bacteria for the effective degradation of BDE-209. In this study, 81.3% of 10 mg L^−1^ BDE-209 could be degraded by the complex bacteria community M (1+2) in 120 h.

In general, temperature directly affected the growth and metabolism of microorganisms and the activity of enzymes, which played key roles in the biodegradation of organic pollutants. Under suitable temperature conditions, bacteria could grow rapidly and produce specific degrading enzymes, thus increasing their ability to degrade organic pollutants. In this study, 30 °C was the optimal temperature for complex bacteria community M (1+2), under which the best degradation effect of BDE-209 was achieved. Moreover, most studies support that the appropriate degradation temperature for BDE-209 is 30 °C, which may be the optimal temperature for degrading enzyme activity involved in BDE-209 biodegradation [32,46,47].

To examine the effect of the pH value on BDE-209 biodegradation by complex bacteria community M (1+2), the solution pH was set at 5.0–9.0. It was found that better BDE-209 degradation results occurred in the range of pH 6.0 to 8.0. This phenomenon was similar to the reports of others. Yu et al. found that *Microbacterium* Y2 showed the highest degradation efficiency of BDE-209 at pH 7.0 [46]. Liu et al. also illustrated that the best effective biodegradation of BDE-209 occurred at pH 7.0 by *Pseudomonas aeruginosa* LY11 [48]. The activity of enzymes depended not only on the ambient temperature but also on pH, and only the optimal pH condition was suitable for an enzyme-catalyzed reaction. In addition, the pH value could cause changes in cell membrane potential and alter the transport of nutrient ions across the membrane, thus affecting the growth of bacteria [46]. A number of surveys have indicated that the environmental pH value occupied an important position in the degradation of organic contaminants because excessive acidity or alkalinity failed to provide a beneficial environment for strain growth and the secretion of degradation enzymes.

The results of BDE-209 degradation under different inoculation volumes showed that degradation efficiency was greatly improved and reached more than 80% when the inoculation volume was higher than 1%. This phenomenon indicated that the degradation ability of complex bacteria community M (1+2) for BDE-209 was obviously inhibited by the poor growth of microorganisms, which could be explained by the fact that the synthesis of degradation enzymes had a certain relationship with the growth of microorganisms. When the inoculation volume was lower, the bacteria per unit volume were under great stress from pollutants, and the bacteria could not secrete enough degradation enzymes in a poor growth environment, which resulted in lower degradation efficiency. On the contrary, the higher inoculation volume could lead to the mass growth of bacteria and further increase the exposure of pollutants to cells and, hence, facilitate BDE-209 degradation.

Under optimized degradation conditions, which were pH 7.0, 30 °C and 15% of inoculation volume, the degradation efficiency of BDE-209 could reach 87.8% at 120 h; however, 64.1% of BDE-209 was already degraded at 6 h, which implied that 73.0% of the total degraded BDE-209 was completed within the initial 6 h. Paliya et al. described the degradation characteristics of strain BDE-S1 for BDE-209, and the study showed that the degradation efficiency was only 33% on day 4, and a 65% degradation efficiency was obtained on day 8 [31]. Similarly, the phenomenon that the obvious degradation of *P. aeruginosa* for BDE-209 occurred between days 3 and 5 was reported by Liu et al. [28]. Consequently, the advantage of complex bacteria community degradation was not only a higher degradation efficiency but also a faster degradation rate than of a single bacterium.

According to previous reports, BDE-209 might undergo a debromination reaction under the action of dehalogenase at first and then be debrominated gradually to produce hepta-BDE [29], penta-BDE [49], tri-BDE [29,49] and diphenyl ether [33,50,51]. Diphenyl ether could be then further degraded under the action of diphenyl 1,2-dioxygenase and diphenyl 2,3-dioxygenase [52,53]. The CYP enzyme could mediate the initial oxidative step of organic pollutant metabolism and the hydroxylation of the deactivated C-H bond, which was deduced to be a common PBDE metabolic path [46]. In our study, phenol and some ring-opening products were detected, which indicated that hydroxylation, ether bond breaking, ring-opening and deprotonation, and other oxidation reactions were major biodegradation mechanisms of BDE-209 by the complex bacteria community M (1+2). Wang et al. also found that debromination, hydroxylation, ether bond breaking and ring-opening reactions mainly occurred in the process of BDE-209 degradation by the aerobic bacterium *Bacillus pumilus* LY2, and intermediate products eventually were mineralized to form low molecular weight compounds [49]. However, (Z)-hex-2-enedioic acid (product i) detected in our study has not been reported up to now. Based on previous work [34,50,52,54], a complex biodegradation pathway for BDE-209 was speculated in this paper. This degradation path indicated that the complex bacteria community M (1+2) could achieve the entire reductive debromination of BDE-209 and the further metabolism of debromination products. The detected four ring-opening products might be completely mineralized to CO_2_ and H_2_O finally through the tricarboxylic acid cycle.

The extracellular and intracellular enzymes secreted by microorganisms play important roles in the organic pollutants’ degradation [47,55]. In this paper, it was proved that intracellular enzymes are the major contributor during BDE-209 degradation. This was consistent with the previous research of Yu et al. [42], in which the intracellular enzymes of *Microbacterium* Y2 dominated the mechanism for BDE-209 biodegradation. However, the contrary phenomenon was also observed. For example, white-rot fungus *Phlebia lindtneri* mainly relied on extracellular enzymes to achieve the effective degradation of BDE-209 [56]. According to our experiments, it is reasonable to deduce that BDE-209 degradation mainly relied on the enzymes existing inside the cells of complex bacteria community M (1+2).

As a common inhibitor of CYP enzymes, PB is often used to examine whether the degradation reaction was catalyzed by CYP enzymes. Previous studies reported that CYP monooxygenases might be involved in the oxidative metabolism of organic compounds [33,57]. Up until now, there has been a lack of knowledge about the qualitative analysis of degradation enzymes and mechanisms responsible for the BDE-209 biodegradation process. In consequence, the present study aimed to investigate the possible role of CYP enzymes during BDE-209 degradation. In this experiment, the obvious inhibition for BDE-209 degradation was observed and the degradation efficiency gradually declined when the concentration of PB gradually increased. The results suggested that cytochrome P450 enzymes played a major role in the degradation of BDE-209 by the complex bacteria community M (1+2). Wei et al. also reported the biotransformation of triphenyl phosphate (TPHP) by *Brevibacillus brevis*, in which the addition of PB markedly decreased the degradation of TPHP [58]. A similar phenomenon was observed during the biodegradation of organic pollutants by *Phanerochaete*. Wang et al. found that PB could affect the hydroxylation of BPA by *Phanerochaete sordida* YK-624 under non-ligninolytic conditions, and the biotransformation activity of BPA was obviously lower in the cultures containing PB [59]. Chen et al. found that CYP450 monooxygenase was significantly upregulated during the degradation of tetrabromobisphenol A (TBBPA) with *Phanerochaete chrysosporium* [60]. These conclusions demonstrated that CYP enzymes were key enzymes in the microbial degradation process of organic contaminants with aromatic ring structures due to their various biocatalytic activities such as oxidation, epoxidation, hydroxylation and demethylation. However, according to previous reports, biphenyl 1, 2-dioxygenase [34,52], catechol 2, 3-dioxygenase [43] and biphenyl 2, 3-dioxygenase [53] might be involved in the biodegradation of BDE-209. Further research will be carried out in this area in the future.

So far, the complex bacterial community used in this study has not yet been applied to the actual environment. Due to the complexity of the environment, it is possible that antagonistic effects of these bacteria appear during the BDE-209 degradation. Maybe the optimization of environmental conditions is one of the better ways to obtain more bacteria in quantity and with higher activity because these BDE-degrading bacteria with high concentration and synergistic effects may become the dominant bacteria in the environment, weakening the antagonistic effects in the bacterial community. At the same time, follow-up monitoring and tracking are also necessary, such as through 16S rRNA microbial community analysis.

## 5. Conclusions

The complex bacteria community of *Brevibacillus* sp. (M1) and *Achromobacter* sp. (M2) with an effective degrading capacity for BDE-209 was studied in this paper. This study reported the BDE-209 biodegradation by the complex bacteria community M (1+2) under aerobic conditions in an aqueous solution for the first time. The degradation efficiency of 10 mg L^−1^ BDE-209 was up to 88.4% after 120 h under optimal biodegradation conditions, which were pH 7.0, 30 °C and 15% of inoculation volume. The identification of metabolic products during BDE-209 degradation suggested that debromination, hydroxylation, deprotonation, the breakage of ether bonds and ring-opening processes were involved in the process of BDE-209 biodegradation. Compared with extracellular crude enzymes, intracellular enzymes were of great importance in the degradation process of BDE-209 by the complex bacteria community M (1+2). The significant decrease in BDE-209 degradation efficiency caused by PB indicated that CYP enzymes probably played a crucial role in the degradation of BDE-209. This study provided an efficient bioremediation method for an environment polluted by BDE-209 using a novel complex bacteria community.

## Figures and Tables

**Figure 1 ijerph-19-17012-f001:**
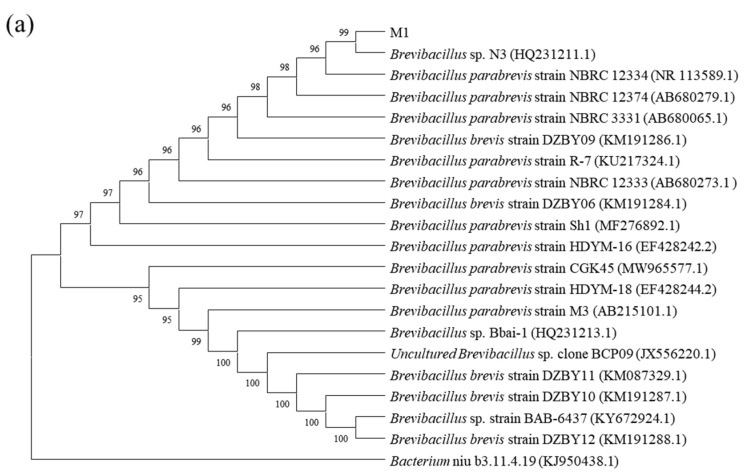
Phylogenetic trees based on 16S rDNA sequence analysis for BDE-209 degrading strains *Brevibacillus* sp. (**a**), *Achromobacter xylosoxidans* (**b**), *Achromobacter xylosoxidans* (**c**) and *Pseudomonas denitrificans* (**d**) which were stored in the China Typical Culture Conservation Center.

**Figure 2 ijerph-19-17012-f002:**
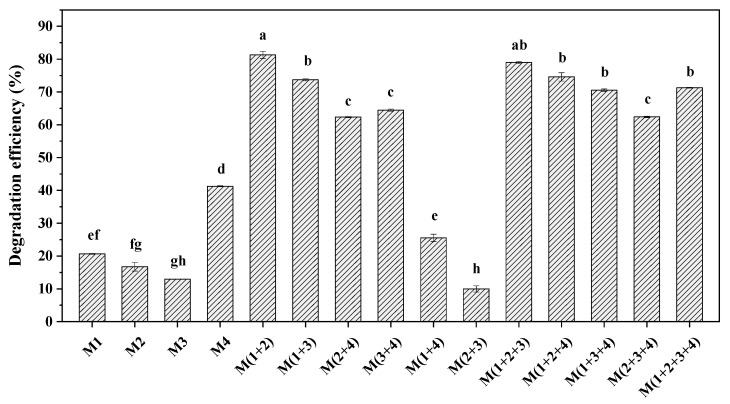
Degradation efficiency of 10 mg L^−1^ BDE-209 by single bacterium and different complex bacteria communities (different lowercase letters indicate statistically significant differences between different treatments (*p* < 0.05)).

**Figure 3 ijerph-19-17012-f003:**
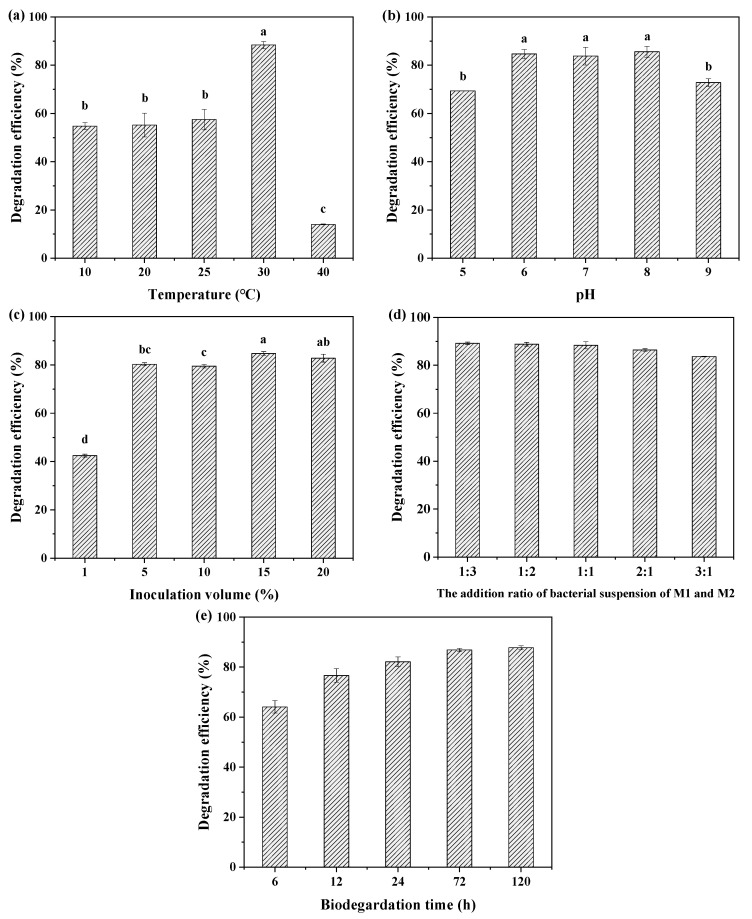
Effects of temperature (**a**), pH (**b**), inoculation volume (**c**), addition ratio of bacterial suspension (**d**) and time (**e**) on BDE-209 degradation by complex bacteria community M (1+2) (different lowercase letters indicate statistically significant differences between different treatments (*p* < 0.05)).

**Figure 4 ijerph-19-17012-f004:**
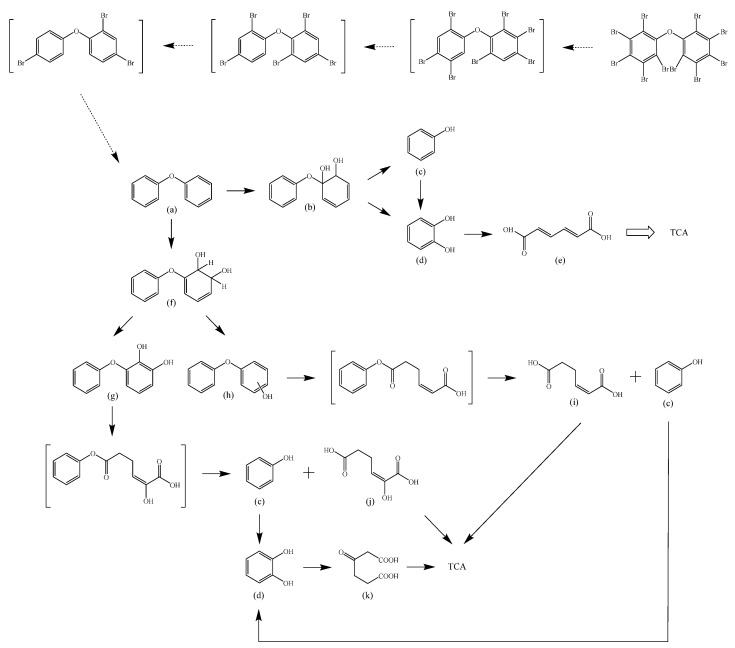
Proposed pathway for BDE-209 biodegradation by complex bacteria community M (1+2).

**Figure 5 ijerph-19-17012-f005:**
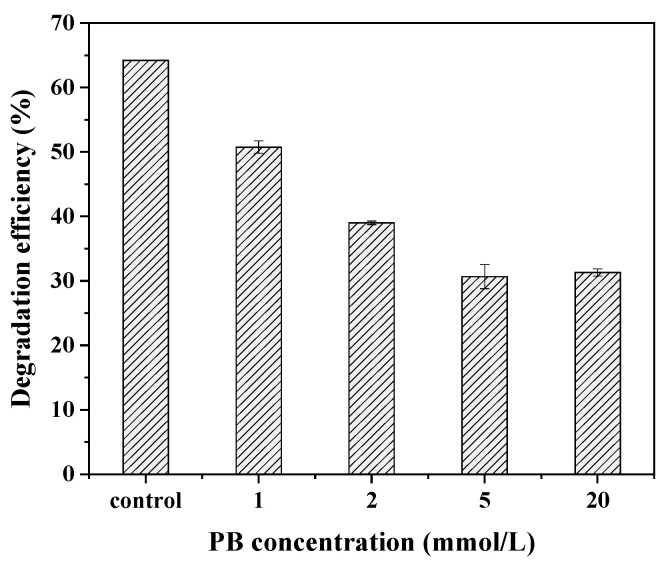
Effect of CYP enzymes inhibitor PB on BDE-209 biodegradation by complex bacteria community M (1+2).

**Table 1 ijerph-19-17012-t001:** The design of optimization experiments for BDE-209 degradation by complex bacteria communities.

	Level	1	2	3	4	5
Single Factor Variable						
Temperature (°C)		10	20	25	30	40
pH value		5	6	7	8	9
Inoculation volume (%)		1	5	10	15	20

**Table 2 ijerph-19-17012-t002:** Metabolites produced by BDE-209 biodegradation by complex bacteria community M (1+2) detected through LC–MS.

Products	Name	Possible Chemical Structure
a	diphenyl ether	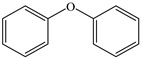
b	1-phenoxycyclohexa-3,5-diene-1,2-diol	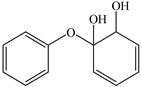
c	phenol	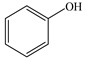
d	pyrocatechol	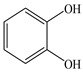
e	(2E,4E)-hexa-2,4-dienedioic acid	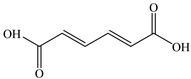
f	2,3-dihydrodiol diphenyl ether	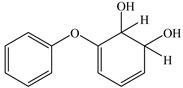
g	2,3-dihydroxydiphenyl ether	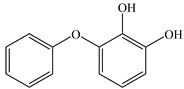
h	monohydroxydiphenyl ether	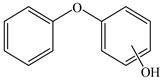
i	(Z)-hex-2-enedioic acid	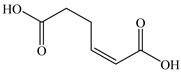
j	(E)-2-hydroxyhex-2-enedioic acid	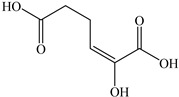
k	3-oxohexanedioic acid	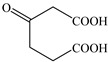

**Table 3 ijerph-19-17012-t003:** Degradation efficiency of BDE-209 by crude enzymes.

	Degradation Efficiency (%)		
	M1	M2	M (1+2)
Extracellular crude enzymes	6.29 ± 0.01	15.99 ± 0.02	20.02 ± 0.01
Intracellular crude enzymes	61.69 ± 0.02	5.08 ± 0.03	69.18 ± 0.06

## Data Availability

The authors declare that all data supporting the findings of this study are available within the article and its Appendix A.

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
