# Peer review of "Aerobic Degradation Characteristics and Mechanism of Decabromodiphenyl Ether (BDE-209) Using Complex Bacteria Communities"

_ijerph, 2022, doi:10.3390/ijerph192417012_

Round 1
Reviewer 1 Report
The authors describe a bacterial community that is able to degrade BDE-209 up to 88%. This study advances the ability to degrade this pollutant and offers insight into the metabolism by analysis of the metabolites of degradation. This reviewer recommends publication after fixing some formatting errors with references.
One minor comment/question for consideration.
Clearly, P450's are involved here as is almost always the case. Can you identify any other enzymes that might be degrading BDE-209 other than P450 and possibly a dehalogenase (maybe non-heme iron)?
Reviewer 2 Report
The manuscript presents the findings of an aerobic degradation study for BDE-209 by a complex bacteria community. The study provides information on the optimal conditions for degradation, enzymes responsible for degradation, and possible degradation products. The manuscript is clear and relevant to the field. However, I recommend the improvement of the manuscript from the points I have listed below.
Major comments:
- In my opinion, one crucial part of these types of degradation studies is the QA/QC part for HPLC and LC-MS analysis. BDE-209 was analyzed by HPLC, what are the method detection limit and limit of quantitation? What is the accuracy and precision of the extraction method used to analyze BDE-209 from samples? The degradation products of BDE-209 were analyzed by LC-MS. As far as I understood, this was a qualitative assessment. Did the authors use the same extraction procedure with BDE-209? How did the authors determine the degradation products, by searching them from the library of LC-MS systems? This needs to be clarified in the methods section. Additionally, it would be good if the authors add the brand of HPLC and LC-MS instruments they used.
- Were all the products listed in the manuscript observed at the end of incubation? In other words, was only the 120th hour sample analyzed for degradation products? Did the authors analyze/search for intermediate products of BDE-209, such as lower brominated PBDE congeners or OH-BDEs? Aerobic degradation also includes debromination mechanism and this would also cause formation of OH-BDEs. Hence it would be better if the authors discuss other possible degradation products that might be observed (if they did not analyze them already in this study).
- Fig. 2: Are there statistically significant differences between M(1+2) & M(1+2+3)? Also, for M(1+2+4) & M(1+3)? It would be better if the authors presented the statistical differences between the results of combined microorganisms.
- Fig. 3 and related discussion in Section 3.3: The differences between the test conditions would be better represented if statistical analyses are conducted. Hence, I would recommend including a statistical comparison between the tested pH, temperature, inoculation volume.
- Can you please give citations to your previous work (Lines 77-79, 99-100)?
- What do the authors mean by “compound microorganism”? I am not very familiar with this terminology, can the authors please explain this term briefly?
- All the bacterial strains were isolated from the activated sludge of a wastewater treatment plant. This means that all the strains appear in the activated sludge system. If there are antagonistic effects of these bacteria for BDE-209 degradation when they co-exist, what should we anticipate to happen in an activated sludge system of a treatment plant or in the environment. How would the authors comment on that? How can we design our treatment systems?
Minor comments:
- Some of the references do not appear in the format of the journal.
- Lines 239-240: From this statement, it is understood that the increase in degradation efficiency is 88.4% compared to others. However, from Fig.3, I understand that 88.4% is the degradation percentage. I recommend rewriting the statement.
Round 2
Reviewer 2 Report
The manuscript has been improved substantially by the revisions made. There are a few points that need further improvement.
-Lines 195-196: How did the authors test the accuracy and precision of BDE-209 extraction? Are the values given here represent analyte recovery? What is the number of samples used for these tests (i.e., n=?)?
-Fig.2: The authors stated that “*” in the figure shows the significant difference between treatments, but there is only one “*” in the figure, which leads to the misunderstanding that all other treatments had no significant difference between them. It would be better if the authors present the statistical differences between the treatments just like they did in Fig. 3.
-The authors stated in their response document that the degradation products were qualitatively detected by comparing their mass spectra to that in the literature studies. It is recommended that they include this statement in Section 3.4.
- The discussion authors did on the reviewer comment of possible antagonistic effects of mixed bacteria could be included in the discussion section of the manuscript.
